# Genetically Engineered Triple *MAPT*-Mutant Human-Induced Pluripotent Stem Cells (N279K, P301L, and E10+16 Mutations) Exhibit Impairments in Mitochondrial Bioenergetics and Dynamics

**DOI:** 10.3390/cells12101385

**Published:** 2023-05-13

**Authors:** Leonora Szabo, Amandine Grimm, Juan Antonio García-León, Catherine M. Verfaillie, Anne Eckert

**Affiliations:** 1Research Cluster Molecular and Cognitive Neurosciences, University of Basel, 4002 Basel, Switzerland; 2Neurobiology Lab for Brain Aging and Mental Health, University Psychiatric Clinics Basel, 4002 Basel, Switzerland; 3Department of Biomedicine, University of Basel, 4055 Basel, Switzerland; 4Departamento Biologia Celular, Genetica y Fisiologia, Instituto de Investigacion Biomedica de Malaga-IBIMA, Facultad de Ciencias, Universidad de Malaga, 29071 Malaga, Spain; 5Centro de Investigacion Biomedica en Red Sobre Enfermedades Neurodegenerativas (CIBERNED), 28031 Madrid, Spain; 6Department of Development and Regeneration, Stem Cell Institute, KU Leuven, 3000 Leuven, Belgium

**Keywords:** tau protein, induced pluripotent stem cells, mitochondria, bioenergetics, oxidative stress, dynamics

## Abstract

Pathological abnormalities in the tau protein give rise to a variety of neurodegenerative diseases, conjointly termed tauopathies. Several tau mutations have been identified in the tau-encoding gene *MAPT*, affecting either the physical properties of tau or resulting in altered tau splicing. At early disease stages, mitochondrial dysfunction was highlighted with mutant tau compromising almost every aspect of mitochondrial function. Additionally, mitochondria have emerged as fundamental regulators of stem cell function. Here, we show that compared to the isogenic wild-type triple *MAPT*-mutant human-induced pluripotent stem cells, bearing the pathogenic N279K, P301L, and E10+16 mutations, exhibit deficits in mitochondrial bioenergetics and present altered parameters linked to the metabolic regulation of mitochondria. Moreover, we demonstrate that the triple tau mutations disturb the cellular redox homeostasis and modify the mitochondrial network morphology and distribution. This study provides the first characterization of disease-associated tau-mediated mitochondrial impairments in an advanced human cellular tau pathology model at early disease stages, ranging from mitochondrial bioenergetics to dynamics. Consequently, comprehending better the influence of dysfunctional mitochondria on the development and differentiation of stem cells and their contribution to disease progression may thus assist in the potential prevention and treatment of tau-related neurodegeneration.

## 1. Introduction

Pathological modifications of the microtubule-associated protein tau are implicated in the pathogenesis of multiple sporadic and familial neurodegenerative diseases, such as frontotemporal dementia, corticobasal degeneration, and progressive supranuclear palsy, which are collectively referred to as tauopathies [1]. Despite presenting clinically heterogenous phenotypes, they are all characterized by an aberrant intraneuronal accumulation of neurofibrillary tangles (NFTs) composed of filamentous hyperphosphorylated tau protein [2]. In particular, the identification of mutations in the tau-encoding gene *MAPT* provided confirmation that tau dysfunction is sufficient to trigger neurodegeneration [3,4,5]. To date, over 50 different mutations have been discovered in *MAPT* [6] and these either affect the physical properties of tau or result in an altered tau splicing. Specifically, the P301L mutation in exon 10 has been shown to disrupt microtubule-binding kinetics, thereby rendering tau more susceptible to hyperphosphorylation, and further promoting the propensity of tau aggregation [7]. In contrast, mutations in intron 10, such as the E10+16 mutation, destabilize the stem-loop structure that regulates the alternative splicing capacity of exon 10, in that way increasing the inclusion of exon 10 and thus enhancing the expression of 4R tau isoforms [4]. Similarly to many intronic mutations, the N279K mutation in exon 10 has been demonstrated to also alter exon 10 splicing, consequently causing a shift in the ratio between 3R and 4R isoforms [8,9]. Of note, these three aforementioned mutations are described as the most prevalent ones observed in frontotemporal dementia with parkinsonism-17, accounting for up to 60% of all cases [10]. However, even within family relatives who carry the same mutation, the phenotypes are wide-ranging [11].

Over the past years, human-induced pluripotent stem cells (iPSCs) have emerged as powerful resources for the development of improved human neuronal models to study the pathogenic mechanisms underlying neurodegenerative diseases [12]. Moreover, they provide an enormous potential for application in regenerative medicine and patient-specific cell and drug therapy [13]. Given that human iPSCs feature the ability to self-renew unlimitedly and possess the capacity to differentiate into any neuronal cell type [14], the resulting neuronal cultures have the advantage of endogenously expressing mutant genes of interest [15]. Indeed, to investigate the molecular processes of tau pathology, iPSCs have been generated from individuals carrying different *MAPT* mutations and subsequently differentiated into mainly cortical neurons, which are known to be affected in tauopathies [16]. However, although various aspects of tauopathy have been recapitulated in these studies, the underlying mechanisms of tau-induced neuronal dysfunction and death remain partially apprehended.

Nevertheless, a growing body of evidence indicates the dysfunction of mitochondria in tauopathies, considering that an insufficiency in mitochondrial function was already discovered at early disease stages, probably appearing even prior to the onset of cognitive impairments [17,18]. Strikingly, disease-associated tau emerges to affect almost every facet of mitochondrial function, as we have reviewed recently [19]. Interestingly, mounting indications highlight the fundamental role of mitochondria in regulating multiple stem cell functions, ranging from stem cell maintenance and survival to stem cell commitment, differentiation, and maturation [20]. Correspondingly, malfunctioning mitochondria are recurrently coupled to various developmental defects, probably due to impairments in early lineage differentiation [21]. Bearing in mind that at the present time, no effective disease-modifying therapies are available for tauopathy patients, the prevention and possible treatment of tau pathology demonstrate an unmet medical need. While the vast majority of studies have focused on predominantly final disease stages, the molecular events giving rise to neuronal loss may begin even earlier [22]. Accordingly, it is of great importance to also explore the early molecular events for a better understanding of the disease progression over time. Thus, iPSCs represent a promising model to study molecular mechanisms of disease-associated tau in early human development. 

In the present study, we examined for the first time the impact of the pathogenic N279K, P301L, and E10+16 mutations on mitochondrial function in human iPSCs. Previously, García-León and colleagues demonstrated that knock-in of these three mutations in the *MAPT* gene reproduced multiple neurodegenerative phenotypes associated with tauopathies in differentiated cortical neurons, including altered tau isoform expression, tau hyperphosphorylation, tau aggregation, impaired neurite outgrowth, and activation of stress response pathways [23]. In that study, the triple tau-mutant iPSCs neither presented tau hyperphosphorylation nor tau aggregation on the iPSCs level yet and exclusively expressed the 3R tau isoform; thus, in the current study, we specifically aimed to investigate the direct effects of the triple tau mutations on mitochondrial performance in the absence of these tau modifications. Here, we report that compared to the isogenic wild-type (iso-WT), triple tau-mutant human iPSCs present deficits in mitochondrial respiration and glycolysis, resulting in the depolarization of the mitochondrial membrane potential (MMP) and the reduced production of adenosine triphosphate (ATP). Moreover, the triple tau mutations disturb the levels of the metabolic redox intermediates nicotinamide adenine dinucleotide (NAD^+^ and NADH), which is coupled with a lessened transcriptional expression of cellular metabolic regulators, leading to decreased mitochondrial biogenesis. Further, we demonstrate that triple tau-mutant human iPSCs exhibit augmented production in mitochondrial reactive oxygen species (ROS), but lowered levels in specific mitochondrial superoxide anion radicals (O_2_^•−^) and cytosolic ROS, due to an enhanced expression of antioxidant enzymes. Besides, the triple tau mutations modify the mitochondrial network morphology, inducing a shift towards a more fragmented state with shorter mitochondria and increased fission, which is paralleled with a perinuclear mitochondrial clustering. In addition, we describe how these alterations in mitochondrial function may be implicated in aberrant stem cell development and differentiation.

## 2. Materials and Methods

### 2.1. Chemicals and Reagents

Tetramethylrhodamine methyl ester perchlorate (TMRM), Hanks’ Balanced Salt Solution (HBSS), 2′,7′-dichlorodihydrofluorescein diacetate (H_2_DCF-DA), dihydrorhodamine 123 (DHR), ethylenediaminetetraacetic acid (EDTA), 3-(4,5-dimethylthiazol-2-yl)-2,5-diphenyltetrazolium bromide (MTT), phenazine ethosulfate (PES), β-nicotinamide adenine dinucleotide hydrate (NAD), β-nicotinamide adenine dinucleotide, reduced disodium salt hydrate (NADH), alcohol dehydrogenase from Saccharomyces cerevisiae (ADH), NP-40, sodium dodecyl sulfate (SDS), formaldehyde, glycine, triton x-100, and bovine serum albumin (BSA) were all purchased from Sigma-Aldrich (St. Louis, MO, USA). Seahorse XFp Cell Mito Stress Test Kit, Seahorse XF Calibrant Solution, Seahorse XF DMEM Assay Medium, pH 7.4, glucose, pyruvate, and glutamine were obtained from Agilent Technologies (Santa Clara, CA, USA). Cellartis DEF-CS 500 Culture System and Cellartis DEF-CS 500 COAT-1 were from Takara Bio (Kusatsu, Shiga, Japan). Phosphate-Buffered Saline (PBS) was acquired from Dominique DUTSCHER SAS (Bernolsheim, France). MitoSOX™ Red Mitochondrial Superoxide Indicator, Tris Ultrapure, and TO-PRO-3 Iodide were purchased from Invitrogen (Waltham, MA, USA). GoScript™ Reverse Transcription Mix, Oligo(dT) and GoTaq^®^ qPCR Master Mix were obtained from Promega (Madison, WI, USA). TrypLE™ Select Enzyme was from Gibco (Waltham, MA, USA), ATPlite 1step Luminescence Assay was from Perkin Elmer (Waltham, MA, USA), Tricine was from Santa Cruz Biotechnology (Dallas, TX, USA), and NaCl was from Merck (Darmstadt, Germany). DC Protein Assay was purchased from Bio-Rad (Hercules, CA, USA), RNeasy Mini Kit from Qiagen (Hilden, Germany), and Vectashield H-1000 Mounting Medium from Vector Laboratories (Newark, CA, USA). Recombinant Alexa Fluor^®^ 555 Anti-TOMM20 antibody was acquired from abcam (Cambridge, UK). All primer sequences were custom-made and purchased from Microsynth (Balgach, Switzerland).

### 2.2. Human iPSCs Culture

Human iPSCs derived from human skin fibroblasts of a non-diseased 24-year-old male donor (iso-WT iPSCs) and triple *MAPT*-mutant human iPSCs bearing the N279K, P301L, and E10+16 mutations (triple tau-mutant iPSCs) were kindly provided by Prof. Catherine M. Verfaillie. The triple tau-mutant iPSCs were generated by introducing the three mutations in and next to exon 10 of the *MAPT* gene into the isogenic control iso-WT iPSCs, using the clustered regularly interspaced short palindromic repeats (CRISPR) FokI and piggyBac transposase technology, as previously described [23]. The iPSCs were maintained under feeder-free conditions and cultured on Cellartis DEF-CS COAT-1-coated plates in Cellartis DEF-CS basal medium, containing the Cellartis DEF-CS additives GF-1 (1:333) and GF-2 (1:1000). The iPSCs were kept in a humidified incubator at 37 °C and 5% CO_2_ with medium replacement every day. The iso-WT iPSCs were passaged twice a week and the triple tau-mutant iPSCs once with TripLE Select, according to the manufacturer’s protocol (Takara). To minimize cell death while passaging or plating, additionally, the Cellartis DEF-CS additive GF-3 was added (1:1000) to the medium.

### 2.3. ATP Levels

The total ATP content was determined using the ATPlite 1step Luminescence Assay following the instructions of the manufacturer. The method measures the production of light, which is formed through the reaction of ATP with luciferin, catalyzed by the enzyme luciferase. The iso-WT and triple tau-mutant iPSCs were plated each with 8–12 replicates into a Cellartis DEF-CS COAT-1-coated white 96-well cell culture plate at a density of 20,000 cells per well. The day after, the wells for the ATP standard curve were prepared, and then 100 µL of ATP substrate solution was added to every well. After incubation in the dark for 2 min under agitation at room temperature, the luminescence was measured using the Cytation 3 Cell Imaging Multi-mode Plate Reader (BioTek). The emitted light was linearly correlated to the ATP concentration and the data were normalized on the protein content.

### 2.4. Determination of MMP

Changes in MMP were measured using the potentiometric fluorescent TMRM as its transmembrane distribution depends on the MMP. The iso-WT and triple tau-mutant iPSCs were plated each with 8–12 replicates into a Cellartis DEF-CS COAT-1-coated black 96-well cell culture plate at a density of 20,000 cells per well. The following day, the iPSCs were loaded in the dark with the dye at a final concentration of 0.4 µM for 30 min under agitation at room temperature. After washing twice with HBSS, the fluorescence signal was detected at 531 nm (excitation)/595 nm (emission) using the Cytation 3 Cell Imaging Multi-mode Plate Reader (BioTek), and the data were normalized on the protein content.

### 2.5. Profiling Mitochondrial Respiration

Key parameters related to mitochondrial respiration were investigated using the Seahorse XF HS Mini Analyzer (Agilent), allowing for the simultaneous real-time measurement of the oxygen consumption rate (OCR) and the extracellular acidification rate (ECAR). The iso-WT and triple tau-mutant iPSCs were plated each with 3 replicates into a Cellartis DEF-CS COAT-1-coated Seahorse XFp Cell Culture Miniplate (Agilent Technologies) at a density of 15,000 cells per well. The following day, the XF Mito Stress Test protocol was performed according to the manufacturer’s instructions. For the measurement, the assay medium consisted of the Seahorse XF DMEM medium, pH 7.4 supplemented with 18 mM glucose, 4 mM pyruvate, and 2 mM L-glutamine. The OCR and ECAR were recorded simultaneously, first under basal conditions, followed by the sequential injection of oligomycin (1.5 µM), carbonyl cyanide-p-trifluoromethoxyphenylhydrazone (FCCP, 1 µM), and a combination of antimycin A (0.5 µM) and rotenone (1 µM). The obtained data were analyzed on the Agilent Seahorse Analytics website, which automatically calculated the bioenergetic parameters, including basal respiration, proton leak, maximal respiration, spare respiratory capacity, non-mitochondrial oxygen consumption, and ATP-production coupled respiration (Table 1). The data were normalized on the protein content.

### 2.6. Detection of ROS Levels

The levels of total cytosolic ROS, total mitochondrial ROS, and the specific level of mitochondrial superoxide anion radicals were assessed using the fluorescent dyes H_2_DCF-DA, DHR, and the Red Mitochondrial Superoxide Anion Indicator (MitoSOX), respectively. The iso-WT and triple tau-mutant iPSCs were plated each with 8–12 replicates into Cellartis DEF-CS COAT-1-coated black 96-well cell culture plates at a density of 20,000 cells per well. The next day, the iPSCs were loaded in the dark with a final concentration of 10 µM of DCF for 20 min, with 10 µM of DHR for 15 min, or with 5 µM of MitoSOX for 2 h under agitation at room temperature. Afterward, the plates were washed twice with HBSS before measuring. During the incubation, H2DCF-DA and DHR are oxidized to dichlorofluorescein (DCF) and cationic rhodamine 123, respectively, generating fluorescent products. The fluorescence signals were detected at 485 nm (excitation)/535 nm (emission) using the Cytation 3 Cell Imaging Multi-mode Plate Reader (BioTek). MitoSOX is oxidized specifically by mitochondrial superoxide, forming a highly fluorescent product, which was detected at 531 nm (excitation)/595 nm (emission). The fluorescence intensities were proportional to cytosolic ROS levels, mitochondrial ROS levels, and superoxide anion radicals in mitochondria. The data were normalized on the protein content.

### 2.7. NAD^+^ and NADH Quantification

To determine the NAD^+^ and NADH concentrations in iPSCs, a spectrophotometric enzyme cycling assay was performed. The method involves the passing of electrons from ethanol through reduced pyridine nucleotides to the final electron acceptor MTT in a PES-coupled reaction, resulting in the formation of a purple precipitate (formazan). The iso-WT and triple tau-mutant iPSCs were plated each into one well of a Cellartis DEF-CS COAT-1-coated 6-well cell culture plate. After reaching 90% confluency, iPSCs were washed once with PBS(-) and detached using TripLE Select. Then, the samples were centrifuged for 5 min at 4 °C at 600 rpm, followed by aspiration of the supernatant. Subsequently, the cell pellets were resuspended in PBS(-), centrifuged again for 5 min at 4 °C at 600 rpm, and the supernatant was aspirated. The cell pellets were then resuspended in protein lysis buffer, and with one part protein, quantification was conducted. The remaining volumes were taken to separately extract NAD^+^ and NADH using an acid–base extraction process (HCL 0.1 M–NaOH 0.1 M). Then, a mix containing equal volumes of 0.1 M Tricine–NaOH buffer, 40 mM EDTA, 4.2 mM MTT, 16.6 mM PES, and 5 M ethanol was added to the samples and incubated for 10 min at 37 °C under agitation. Enzyme cycling was initiated by the subsequent addition of 100 U ADH per well and carried out for 1 h at 37 °C under agitation. The enzyme reaction was terminated with the addition of 6 M NaCl to accelerate the precipitation of formazan. Afterward, ethanol 96% (*v*/*v*) was added for solubilization, and the absorbance was quantitatively measured at 595 nm using the Cytation 3 Cell Imaging Multi-mode Plate Reader (BioTek).

### 2.8. Protein Content Quantification

To assess the protein content of samples, the DC Protein Assay was used with BSA as standard, according to the manufacturer’s instructions. For sample preparation, the cells were dissolved in protein lysis buffer, containing 150 mM Tris Ultrapure, 150 mM NaCl, 1% NP-40, 0.1% SDS, and 2 mM EDTA. The absorbance of samples was measured at 690 nm using the Cytation 3 Cell Imaging Multi-mode Plate Reader (BioTek). The protein content of samples was used to normalize the obtained data from experiments.

### 2.9. Total RNA Isolation, cDNA Synthesis, and Quantitative Real-Time PCR

To characterize the effects of the triple tau mutations on mitochondrial function, the expression levels of genes coding for different proteins were investigated. The iso-WT and triple tau-mutant iPSCs were plated each into three wells of a Cellartis DEF-CS COAT-1-coated 6-well cell culture plate. After reaching 80% confluency, iPSCs were washed once with PBS(-), lysed with RLT buffer, and subsequently homogenized for at least 30 s at 400 min^−1^ using the rotor-stator homogenizer Potter S (B. Braun Biotech). Total RNA was isolated from homogenized cell lysates using the RNeasy Mini Kit as described by the manufacturer. RNA concentrations were quantified with the Cytation 3 Cell Imaging Multi-mode Plate Reader (BioTek). To synthesize cDNA, the volume containing 1 µg of RNA was reverse-transcribed using the GoScript™ Reverse Transcription Mix, Oligo(dT) according to the manufacturer’s instructions. After reverse transcription, all cDNA samples were diluted 1:5 in nuclease-free water. For the amplification of cDNA samples, the GoTaq^®^ qPCR Master Mix was used with the addition of 300 nM CXR reference dye. The final concentrations of the forward and reverse primers (Table 2) were optimized for each primer combination, ranging from 200 to 900 nM. Quantitative real-time PCR reactions were performed in technical duplicate using the StepOnePlusTM Real-Time PCR System (Applied Biosystems, Waltham, MA, USA) under the following conditions: for 2 min at 95 °C, for 3 s at 95 °C followed by 30 s at 60 °C, for a total of 40 cycles. GAPDH was used as an endogenous reference and the final quantification of relative fold gene expression was calculated based on the 2^−ΔΔCt^ method.

### 2.10. Mitochondrial Morphology and Distribution

To visualize and quantify the influence of the triple tau mutations on mitochondrial morphology and distribution, iPSCs were immuno-stained with the mitochondrial marker translocase of the outer mitochondrial membrane complex subunit 20 (TOMM20). Briefly, the iso-WT and triple tau-mutant iPSCs were plated in 12-well cell culture plates on Cellartis DEF-CS COAT-1-coated coverslips at a density of 80,000 cells per well. The following day, iPSCs were fixed in 10% formaldehyde for 10 min at room temperature and the reaction was stopped by the addition of 1 M glycine in PBS(-), serving as a quenching agent. Then, permeabilization was conducted with 0.1% Triton X-100 in PBS(+) for 15 min at room temperature under agitation. Blocking was performed with 2% BSA in PBS(+) for 1 h at room temperature under agitation. For mitochondrial staining, coverslips were incubated with the conjugated Alexa Fluor^®^ 555 rabbit anti-TOMM20 antibody, diluted 1:75 in 1% BSA in PBS(+), for 2 h at room temperature in a humidity chamber. For subsequent nuclear staining, coverslips were incubated with TO-PRO-3, diluted 1:400 in 1% BSA in PBS(+), for 30 min at 37 °C in a humidity chamber. Lastly, coverslips were mounted on glass slides with the Vectashield H-1000 mounting medium and sealed with nail polish.

The acquisition of confocal images was conducted using an inverted confocal microscope (Leica Microsystems TCS SPE DMI4000) connected to an external light source for enhanced fluorescence imaging (Leica EL6000) with the Leica Application Suite Advanced Fluorescence (Leica LAS AF) software (version 2.5.1.6757). The images were taken using an HCX PL APO oil objective with a magnification of 63 × 1.10–0.60, and pinhole settings were selected so that each cell was axially entirely present within the confocal volume. Z-stacks were obtained with 10 steps with a z-volume of 3.021 µm. The acquisition settings were kept constant during imaging. The TOMM20 Alexa Fluor^®^ 555 signal was detected using a 532 nm laser with 30% transmission, a gain of 700, and an emission bandwidth of 555–595 nm. The TO-PRO-3 signal was detected using a 635 nm laser with 30% transmission, a gain of 700, and an emission bandwidth of 654–800 nm. The images were collected on four independent occasions with 15 cells per group. 

Changes in mitochondrial shape were quantified with an automated image processing and morphometry macro using the FIJI software (version v1.53c.) as described earlier [24]. In brief, the z-projection of z-stacks was generated for the images using the maximum intensity projection type. Then, the background was subtracted (rolling ball radius 50 pixels), and uneven labeling of mitochondria was improved through local contrast enhancement using contrast-limited adaptive histogram equalization (“CLAHE”). For the segmentation of mitochondria, the “Tubeness” filter was applied. Then, an automated threshold was set and the “Analyze Particles” command was run to determine the area and the perimeter of individual mitochondria. To measure the mitochondrial length, the “Skeletonize” function was applied. The measurements were limited to a region of interest, representing the area of the whole cell. The average metrics generated by the morphometry macro were calculated from the area, perimeter, the major and minor axis of an elliptical fit of the binary particles, and the area after skeletonizing the binary particles. While the area^2^ measures the average size of mitochondria, the form factor (or mitochondrial elongation) describes the particle’s shape complexity as the inverse of circularity and is particularly reliable for well-separated mitochondria. The area-weighted form factor is a variant of the form factor, providing more credible results in cases where highly elongated mitochondria overlap. The aspect ratio is defined as the ratio of the major and the minor axis, which is independent of the area and perimeter. The length reports the mitochondrial length or elongation in units of pixels, after reducing mitochondria to single-pixel-wide shapes. To analyze the distribution of mitochondria within the iPSCs, the radial profile plugin was used in FIJI. This algorithm measured the radial fluorescence of the TOMM20 signal from the center of the nucleus toward the plasma membrane in a full-angle mode (0–360°).

### 2.11. Statistical Analysis

Data are presented as the mean ± SEM. Statistical analyses and data presentation were performed using Graph Pad Prism 9 (version 9.3.1). Values were normalized on the mean of the iso-WT group (=100%) in each individual experiment and the normalized data of each individual experiment were pooled for subsequent statistical analyses and data presentation. Normal distribution of the data was assessed using the Shapiro–Wilk test, and the Student’s unpaired *t*-test was used for statistical comparisons between the iso-WT and the triple tau-mutant iPSCs. *p*-values < 0.05 were considered statistically significant. Statistical parameters can be found in the figure legends.

## 3. Results

### 3.1. Triple Tau-Mutant iPSCs Display Mitochondrial Bioenergetic Deficits

Previous data of our group reported a negative impact of P301L mutant tau on mitochondrial bioenergetics in human-derived neuroblastoma cells (SH-SY5Y), presenting a decrease in mitochondrial respiration, reduced ATP levels, and a lowered MMP [25,26,27]. Based on these observations, in the present study, we evaluated the bioenergetic profile of triple *MAPT*-mutant human iPSCs bearing the N279K, P301L, and E10+16 mutations compared to iso-WT iPSCs. To address whether the triple tau mutations affect the efficiency of mitochondrial respiration and cellular bioenergetics, we first conducted the Seahorse XF Cell Mito Stress Test using the Seahorse XF HS Mini Analyzer. Specifically, we simultaneously monitored in real time the OCR (Figure 1a), an indicator of mitochondrial oxidative phosphorylation (OXPHOS), as well as the ECAR (Figure 1b), an indicator of glycolysis. We observed a significant decrease in the OCR (Figure 1c) and an even more striking decline in the ECAR (Figure 1d) of triple tau-mutant iPSCs compared to iso-WT iPSCs under basal conditions. Regarding the calculated bioenergetic parameters (Figure 1g,h), triple tau-mutant iPSCs showed a significant reduction in basal respiration, proton leak, maximal respiration, spare respiratory capacity, non-mitochondrial oxygen consumption, ATP-production coupled respiration, coupling efficiency, and spare respiratory efficiency, when compared to iso-WT iPSCs, indicating a profound mitochondrial metabolic impairment. 

In view that ATP is not only the end product of mainly OXPHOS but also of glycolysis, it thus can be used as a general marker of mitochondrial function and cellular viability [28]. However, given that the Seahorse assay indirectly measures ATP production by inhibiting mitochondrial complex V, we next determined the actual ATP levels in iPSCs (Figure 1e). Consistent with the Seahorse-derived data, we found a significant decrease in ATP concentration in triple tau-mutant iPSCs compared to iso-WT iPSCs. Further, the mitochondrial complex V utilizes the proton motive force to catalyze the synthesis of ATP, which in turn depends on the proton gradient generated by complexes I, III, and IV, ultimately forming an electrochemical potential, the MMP [29]. Therefore, we wanted to verify whether the decrease in ATP levels was directly linked to reduced mitochondrial activity (Figure 1f). Indeed, triple tau-mutant iPSCs revealed a significant drop in MMP compared to the one of iso-WT iPSCs. 

Taken together, these data demonstrate that the triple tau mutations exhibit a negative impact on mitochondrial bioenergetics in iPSCs, leading on the one hand to a mitochondrial respiration deficiency, and on the other hand to defective glycolysis. Consequently, triple tau-mutant iPSCs presented a depletion in ATP, which was paralleled by a depolarization of the MMP.

### 3.2. Triple Tau Mutations Impair Parameters Linked to the Metabolic Regulation of Mitochondrial Function in iPSCs

To better understand our findings related to the altered metabolic features of triple tau-mutant iPSCs, we next investigated relevant metabolic intermediates of mitochondrial metabolism. Due to the ability to transfer electrons, NAD serves as an essential cofactor that mediates various redox reactions in cellular energy metabolism [30]. In metabolic processes, including glycolysis, the oxidized form NAD^+^ is reduced into NADH, whereas reduced NADH is oxidized to NAD^+^, thereby acting as a central electron donor in OXPHOS. In the end, both pathways drive the generation of ATP [31]. To quantify the NAD^+^ and NADH metabolite levels, we used an acid–base extraction method and performed a spectrophotometric enzyme cycling assay. Intriguingly, we detected a significant increase in the NAD^+^ concentration of triple tau-mutant iPSCs compared to iso-WT iPSCs (Figure 2a). Conversely, the NADH concentration in triple tau-mutant iPSCs was significantly decreased when compared to iso-WT iPSCs (Figure 2b). Besides, the ratio between oxidized NAD^+^ and reduced NADH plays a crucial role in the maintenance of balanced metabolic redox homeostasis [32]. Hence, the calculated NAD^+^/NADH ratio was significantly augmented in triple tau-mutant iPSCs versus iso-WT iPSCs (Figure 2c).

To gain more insight into possible mechanisms underlying the effects of the triple *MAPT* mutations on abnormal metabolic regulation, we subsequently assessed the expression levels of key genes involved in NAD metabolism and mitochondrial biogenesis (Figure 2d). In addition to cellular bioenergetics, NAD^+^ operates also as a substrate for NAD^+^-consuming enzymes, including sirtuins that belong to the family of NAD^+^-dependent deacetylases [33] and play a pivotal role in the regulation of energy metabolism and mitochondrial function [34]. For this reason, we first examined the gene expression of nuclear sirtuins 1 (SIRT1) and mitochondrial sirtuins 3 (SIRT3). Remarkably, we observed that SIRT1 was significantly downregulated, while SIRT3 was significantly upregulated in triple tau-mutant iPSCs compared to iso-WT iPSCs. Further, SIRT1 deacetylates and positively regulates peroxisome proliferator-activated receptor-γ co-activator-1α (PGC-1α), which is a transcriptional co-activator, functioning as a master regulator of mitochondrial biogenesis and of numerous metabolic processes [35]. Moreover, studies found that in mitochondria, SIRT1 and PGC-1 α also interact with mitochondrial transcription factor A (TFAM) [36]. In accordance with downregulated SIRT1, we detected a significantly diminished PGC-1α and TFAM expression in triple tau-mutant iPSCs compared to iso-WT iPSCs. Because estrogen-related receptor alpha (ERR-α) and nuclear factor erythroid 2-related factor 2 (NRF2) are supposed to activate SIRT3 expression in response to mitochondrial stress [37,38], we next evaluated their expression levels. Compared to iso-WT iPSCs, ERR-α and NRF2 gene products were significantly upregulated in triple tau-mutant iPSCs, consistent with the upregulation of SIRT3.

Collectively, our data indicate an impairment in the metabolic redox potential control in triple tau-mutant iPSCs, causing disturbances in NAD^+^ and NADH levels with a shift towards a highly oxidized state. Although triple tau-mutant iPSCs presented an elevated NAD^+^ concentration, the gene expression level of SIRT1 was drastically reduced, which, however, is in line with the paralleled diminution of PGC-1α and TFAM. Nevertheless, consistent with the observed increase in NAD^+^ metabolites in triple tau-mutant iPSCs, SIRT3, ERR-α, and NRF2 showed an upregulated expression.

### 3.3. Triple Tau-Mutant iPSCs Exhibit a Disturbed Cellular Redox Homeostasis

Although mitochondria remain the main actors in cellular energy production, they also account for the primary source of ROS formation. In fact, mitochondrial ROS mainly derive from the mitochondrial electron transport chain activity as inevitable by-products. During mitochondrial respiration, a leakage of electrons predominantly from complex I and III leads to the partial reduction of oxygen, thereby forming O_2_^•−^, which subsequently can be converted into other ROS, such as hydrogen peroxide (H_2_O_2_) and the highly reactive hydroxyl radical (^•^OH) [39]. Considering that triple tau-mutant iPSCs exhibited deficits in mitochondrial respiration, we measured the levels of total cytosolic ROS, total mitochondrial ROS, and the specific level of O_2_^•−^ using the fluorescent dyes H_2_DCF-DA, DHR, and MitoSOX, respectively. Interestingly, while we detected a significant decrease in cytosolic ROS levels (Figure 3a), mitochondrial ROS levels (Figure 3b) were significantly elevated in triple tau-mutant iPSCs compared to iso-WT iPSCs. Further to our surprise, we observed a significant reduction in O_2_^•−^ levels (Figure 3c) in triple tau-mutant iPSCs when compared to iso-WT iPSCs.

Given that cells possess efficient antioxidant mechanisms to scavenge ROS to non-toxic forms and thus avoid exceeding ROS levels [40], we next investigated the expression levels of genes coding for key antioxidant enzymes (Figure 3d). In general, detoxification primarily comprises the enzymatic activity of superoxide dismutases (SODs), notably SOD1 and SOD2, that convert toxic O_2_^•−^ into H_2_O_2_ and oxygen (O_2_) [41]. Following conversion, catalase (CAT) and glutathione peroxidase 1 (GPX-1) further reduce H_2_O_2_ into water (H_2_O) [42]. Strikingly, we found that all genes involved in the antioxidant defense system were significantly upregulated in triple tau-mutant iPSCs compared to iso-WT iPSCs. In particular, SOD1 and CAT presented the highest overexpression, implying an exceptionally efficient detoxification. 

Altogether, these results revealed that in triple tau-mutant iPSCs, the cellular redox homeostasis is disrupted, resulting in the first place in an increased production of total mitochondrial ROS. However, to protect mitochondria against potential oxidative damage, the gene expression levels of SOD1 and SOD2 were markedly augmented, leading to an effective dismutation of toxic O_2_^•−^. Consequently, triple tau-mutant iPSCs demonstrated decreased mitochondrial O_2_^•−^ levels. In the same way, CAT and GPX-1 displayed an upregulated expression to further increase the antioxidant activity, eventually causing a reduction in total cytosolic ROS levels.

### 3.4. Triple Tau Mutations Modify the Mitochondrial Network Morphology and Distribution in iPSCs

Alterations in the bioenergetic status of mitochondria are often coupled with adaptations of their morphology in response to changes in cellular energy requirements [43]. Indeed, mitochondria are considered as highly dynamic organelles that form a remarkably complex interconnected network, cooperatively acting in a coordinated manner [44]. Accordingly, to maintain a homogeneous and functional mitochondrial population, mitochondria change their shape by continuous rounds of fusion and fission. While increased fusion generates elongated mitochondria with a tubular morphology, enhanced fission promotes mitochondrial fragmentation and isolation [45]. Since triple tau-mutant iPSCs showed bioenergetic deficits, we hence studied the impact of the triple tau mutations on the mitochondrial network morphology using confocal microscopy. To visualize mitochondria and the nucleus, we performed immuno-staining with the mitochondrial markers TOMM20 and TO-PRO-3, respectively, followed by automated image processing and morphometry macro using the FIJI software (version v1.53c.) for the actual quantification of mitochondrial shape (Figure 4a). We detected a significant decrease in all mitochondrial shape descriptors, namely the area^2^ (Figure 4b), the form factor (Figure 4c), the area-weighted form factor (Figure 4d), and the aspect ratio (Figure 4e) in triple tau-mutant iPSCs compared to iso-WT iPSCs. Further, we observed an additional significant reduction in mitochondrial length (Figure 4f) in triple tau-mutant iPSCs versus iso-WT iPSCs, indicating either lessened fusion or augmented fission events. Moreover, as an altered cytoplasmic distribution of mitochondria can severely impact cellular metabolism, we also investigated the mitochondrial distribution within iPSCs using the radial profile plugin in FIJI, which measured the radial fluorescence of the TOMM20 signal. Interestingly, we found a significant increase in the abundance of mitochondria around the nucleus (Figure 4g) in triple tau-mutant iPSCs when compared to iso-WT iPSCs, suggesting that the triple tau mutations not only affect mitochondrial morphology but also the distribution of mitochondria.

Considering that fusion and fission activities can be regulated on a transcriptional level and to dissect more in detail our latest observations, we next examined the expression levels of key genes involved in mitochondrial dynamics (Figure 4h). Mechanistically, mitochondrial fusion requires the actions of three large dynamin-related guanosine triphosphatases (GTPases). While mitofusin 1 (MFN1) and 2 (MFN2) mediate the fusion of the outer mitochondrial membrane, optic atrophy 1 (OPA1) facilitates the fusion of the inner mitochondrial membrane [46]. Surprisingly, MFN1 and MFN2 were significantly upregulated in triple tau-mutant iPSCs compared to iso-WT iPSCs, whereas OPA1 was significantly downregulated. On the contrary, mitochondrial fission is primarily orchestrated by the cytosolic dynamin-related protein 1 (DRP1), which is another GTPase, and additional assisting proteins, such as the mitochondrial fission 1 protein (FIS1). Compared to iso-WT iPSCs, DRP1 and FIS1 presented a pronounced significant overexpression in triple tau-mutant iPSCs, supporting our confocal imaging results that implied increased fission. 

Overall, our results demonstrated that the presence of triple tau mutations in iPSCs alters both mitochondrial dynamics and distribution. As a result, mitochondria become shorter or less elongated and shift their morphology towards a more fragmented state, which is paralleled with a perinuclear clustering. Correspondingly, the gene expression levels of DRP1 and FIS1 were substantially elevated, while that of OPA1 was reduced, confirming further the increase in mitochondrial fission.

## 4. Discussion

Given that mitochondrial dysfunction in tauopathies was identified already at early disease stages, occurring even before the onset of cognitive deficits [17], recent publications have focused on studying the relationship between tau and mitochondria in human iPSC-derived neurons from patients bearing different *MAPT* mutations. However, these studies mainly draw attention to alterations in axonal transport, vesicle trafficking, calcium flux, or neurite outgrowth in fully differentiated cortical neurons [16]. Hence, the characterization of mitochondrial impairments at the human iPSCs level has not yet been investigated. Human iPSCs possess the unique ability to differentiate into any cell type of the body [47]. In addition, mounting evidence now indicates that mitochondria have emerged as central regulators of stem cell function, acting as signaling centers. Thus, it can be understood that mitochondrial dysfunction might cause damaging effects on stem cell development, which in turn may contribute to cognitive processes in the adult brain [20]. In the present study, we used genetically engineered triple *MAPT*-mutant human iPSCs, carrying the N279K, P301L, and E10+16 mutations, to explore mitochondrial bioenergetics and dynamics on the iPSC level for the first time. We have found that the triple *MAPT* mutations negatively affected mitochondrial bioenergetics, altered parameters linked to the metabolic regulation of mitochondria, disturbed cellular redox homeostasis, and modified mitochondrial network morphology and distribution (Figure 5).

To meet the cellular energy requirements, ATP is mainly produced via two pathways, which include glycolysis and OXPHOS. Cytosolic aerobic glycolysis not only metabolizes glucose to pyruvate to yield ATP, but also reduces oxidized NAD^+^ to generate NADH, which serves as an important electron carrier. Subsequently, pyruvate enters mitochondria and is converted into acetyl-CoA, which reaches the tricarboxylic acid (TCA) cycle within the mitochondrial matrix, ultimately resulting in the formation of additional NADH out of NAD^+^ and FADH_2_ from FAD [48]. During OXPHOS, the reducing equivalents NADH and FADH_2_ consequently provide electrons to different respiratory chain complexes of the electron transport chain (ETC), allowing them to pump protons out of the mitochondrial matrix to the intermembrane space, thereby producing a proton gradient that establishes the MMP. As a result, this proton motive force is used to drive the synthesis of ATP [29]. Previous studies had shown that P301L mutant tau impairs mitochondrial bioenergetics in vivo and in vitro. In particular, investigations performed in pR5 mice reported an age-dependent reduction in mitochondrial respiration, mitochondrial complex I activity, and ATP levels [49,50]. Similar findings were demonstrated in SH-SY5Y cells [25], where additionally, a decreased MMP was presented [26,27]. In line with this, our findings in triple tau-mutant iPSCs confirm these observations, revealing a deficiency in mitochondrial respiration together with a depolarization of the MMP, finally leading to ATP depletion. On the other hand, a study in iPSCs-derived neurons carrying the E10+16 tau mutation reported that despite the reduction in ATP production via OXPHOS, the neurons presented similar total ATP levels, possibly due to an increase in aerobic glycolysis [51]. In contrast, triple tau-mutant iPSCs exhibited deficits in glycolysis, and thus, no attempt to metabolically switch on glycolysis occurred to maintain ATP production. Nevertheless, this is hardly surprising as stem cells primarily rely on aerobic glycolysis to generate ATP and intermediates, which in turn are essential for maintaining stem cell properties such as rapid cell proliferation and pluripotency [13]. Regardless of the relatively low contribution of OXPHOS for ATP production, several studies proved that although stem cell mitochondria possess the capacity to promote ATP generation via OXPHOS, it rather seems that they are actively repressed through several mechanisms to do so [20]. Likewise, it was demonstrated that iPSCs are not only able to consume oxygen at maximal capacity with functional respiratory complexes at rates similar to differentiated cells, but also express uncoupling protein 2 (UCP2), which maintains the MMP while suppressing OXPHOS [52]. Although iPSCs mainly depend on glycolysis, our results not only show defective glycolysis with reduced ATP levels in the presence of the triple tau mutations, but also detect already an impairment in mitochondrial respiration coupled with a decreased MMP. Considering that during differentiation into neurons OXPHOS, ATP levels and MMP gradually increase [44], blocking the mitochondrial energy production might lead to a compromised differentiation capability, maturation, and survival. Indeed, García-León et al. reported that the triple *MAPT* mutations resulted in aberrant differentiation into cortical neurons and progressive cell loss [23]. 

As mentioned above, cellular energy metabolism involves relevant metabolic redox intermediates, such as NAD^+^ and NADH, which play crucial roles in modulating numerous biological events and are often considered as reflections of the metabolic state. Given that a decline in NAD^+^ levels in the aging human brain was described in multiple studies and has been associated with altered mitochondrial function leading to neurodegeneration [53,54], we further assessed the NAD^+^ and NADH metabolite levels in our model. Contrary to our expectations, our findings in triple tau-mutant iPSCs revealed an elevated NAD^+^ pool, paralleled with a reduced NADH pool, resulting in an increased NAD^+^/NADH ratio. These data differ from previous studies in models of tauopathy describing the inhibition of mitochondrial complex I activity [25,49,50], which would, in that context, lead to reduced oxidation of NADH into NAD^+^ and therefore to lower NAD^+^ levels and higher NADH levels with a decreased NAD^+^/NADH ratio [55]. However, these discrepancies might be explained by differences in the experimental model in relation to the type of tau mutation used. Nevertheless, in support of our findings, a study in iPSCs-derived neurons bearing the E10+16 tau mutation found a lowered NADH pool in mitochondria, suggesting that the inhibition of the mitochondrial complex I activity may be attributed to a lessened availability of NADH for complex I [51]. Although this may be true, the question still remains why triple tau-mutant iPSCs display higher levels of NAD^+^. A possible explanation could involve the observed diminution in glycolysis, as within this pathway, NAD^+^ is normally reduced to NADH. Consequently, lowering the NAD^+^ consumption could therefore be compatible with the detected accumulation of NAD^+^ molecules. Besides, the contributions of other pathways must be unraveled, since not only inhibitions of mitochondrial β-oxidation and the TCA cycle could similarly promote additional NAD^+^ accumulation, but also augmentations in lactic acid generation from pyruvate and desaturation of polyunsaturated fatty acids (PUFAs) [31]. Of note, NAD^+^ can additionally be synthesized de novo or recycled via the NAD^+^ salvage pathway [30]. 

Importantly, NAD^+^ also acts as a co-substrate for various NAD^+^-consuming enzymes [54]; thus, a decreased catabolism could likewise result in increased NAD^+^ levels. Among these enzymes, sirtuins, a family of NAD^+^-dependent protein deacetylases, are associated with the regulation of cellular energy metabolism [34]. Intriguingly, our study revealed that nuclear SIRT1 was downregulated in triple tau-mutant iPSCs, which is consistent with the findings reported in human tauopathy brains, where reduced SIRT1 levels were detected and negatively correlated with the amount of hyperphosphorylated tau aggregates [56]. Comparatively, a study demonstrated that SIRT1-induced deacetylation of acetylated tau allows the degradation of hyperphosphorylated tau species, hence potentially lowering the formation of NFTs [57]. Furthermore, it has been described that SIRT1 activates PGC-1α via deacetylation, thereby regulating mitochondrial biogenesis and metabolism [58]. In addition, SIRT1 and PGC-1α are able to localize inside mitochondria to interact with TFAM [36]. Supporting our data concerning SIRT1, triple tau-mutant iPSCs presented a downregulation of PGC-1α and TFAM. These results are in complete agreement with findings in P301L tau transgenic mice and P301L mutant HT22 cells, which showed a reduction in mitochondrial biogenesis with decreased expressions of PGC-1α and TFAM [59,60,61,62]. Given these points, the reductions in SIRT1 activity and its downstream effectors PGC-1α and TFAM seem to contribute directly to the decline in mitochondrial function by deleteriously affecting mitochondrial biogenesis and metabolic homeostasis, consequently implicating the importance of the neuroprotective role of SIRT1. Equally fundamental, SIRT1 operates as a multifaceted coordinator in stem cell function, being critically involved in the maintenance and development of stem cells through the regulation of pluripotency factors, energy metabolism, epigenetics, and redox homeostasis [63]. Accordingly, pathology-provoked alterations in its expression may hence disturb various signaling processes that are required for proper stem cell commitment. Interestingly, using human iPSCs with TFAM knockout investigations demonstrated that TFAM deletion caused a proliferation arrest, contributing to impaired self-renewal and ultimately resulting in a severe lineage differentiation defect [21]. 

Changes in the NAD^+^/NADH ratio also affect the activity of other sirtuins, such as mitochondrial SIRT3, which acts as a cellular metabolic sensor to sustain multiple aspects linked to mitochondrial metabolism and function. Moreover, both NRF2 and ERR-α bind to distinct sites of the SIRT3 transcriptional start site and are suggested to activate SIRT3 expression to control the cellular redox status [34,38]. In our study, we detected an upregulated expression of SIRT3, NRF2, and ERR-α in triple tau-mutant iPSCs. Contrary to our findings, a reduced SIRT3 expression was observed in postmortem brain slices of Alzheimer’s disease (AD) patients, which was paralleled with increased tau acetylation levels [64]. In the same way, previous studies in P301L tau transgenic mice and P301L mutant HT22 cells reported a diminished expression of NRF2 [59,60,61,62]. Nonetheless, the upregulation of SIRT3 and NRF2 in our model may represent an endogenous neuroprotective response occurring early in disease pathogenesis that might then be lost during further disease progression. Consistent with this hypothesis is the finding that glycogen synthase kinase-3β (GSK-3β) activity inhibits NRF2 [65]. Notably, GSK-3β activation has been implicated in tau pathology, leading to heightened levels and the accumulation of NFTs composed of hyperphosphorylated tau species [66]. In fact, in an AD mouse model, it was shown that GSK-3β suppression was able to increase NRF2, thereby reducing tau hyperphosphorylation [65]. Consequently, one might presume that during disease progression, the activity of GSK-3β increases, which subsequently causes a decline in NRF2 expression. 

Another key point to consider concerns the involvement of NRF2 in driving vital aspects of stem cell survival and function. Specifically, studies have indicated that NRF2 serves as a fundamental regulator of the cellular metabolic state and redox profile, and in that way, controlling stem cell self-renewal, proliferation, and differentiation [67]. Because enhanced NRF2 expression was shown to upregulate antioxidant enzyme genes [68], the high levels of NRF2 in triple tau-mutant iPSCs may thus be seen as an attempt to modulate the oxidative stress response. Comparatively, the NAD^+^-dependent enzyme SIRT3 acts as an important modulator of the cellular redox state via deacetylation and successive activation of enzymes associated with the antioxidant defense system [69]. In general, ROS are formed through a variety of mechanisms; however, the majority of cellular ROS production originates from mitochondria as unavoidable by-products through the activity of the ETC [70]. Accordingly, the production and detoxification of ROS are balanced under normal conditions, since in many biological processes, physiologically low levels of ROS are required as signaling molecules. Nevertheless, increased ROS formation and/or decreased antioxidant capacity can evoke oxidative stress, which consecutively damages DNA, proteins, and membranes and mediates lipid peroxidation [71,72]. Therefore, to impede exceeding ROS levels and oxidative injury, cells scavenge ROS to non-toxic forms with efficient antioxidative defense mechanisms [40]. Regarding the effects of the triple tau mutations on the antioxidant defense system, our data revealed striking changes. Given that SIRT3 overexpression activates SOD2 and CAT [73,74], the upregulation of SIRT3 and NRF2 in triple tau-mutant iPSCs is compatible with the augmented expression levels of the antioxidant enzymes SOD1, SOD2, CAT, and GPX-1, supporting an induction of the oxidative stress defense to protect against possible oxidative damage. In line with this, García-León et al. reported that the triple tau-mutant neuronal progeny displayed considerably elevated levels of oxidative stress response markers, including NRF2, SOD1, and CAT [23]. Furthermore, previous studies in different animal and cellular models with *MAPT* mutations have indicated oxidative stress-induced mitochondrial dysfunction as a causative factor in tau pathology. Particularly, these investigations described higher levels of cytosolic ROS, mitochondrial ROS, and specific O_2_^•−^ [49,50,51,59,75]. On the contrary, our findings in triple tau-mutant iPSCs deviate slightly from these observations. In fact, we found comparably elevated mitochondrial ROS levels, but decreased total cytosolic ROS and specific O_2_^•−^ levels. Nevertheless, these differences may be due to the special role of ROS in regulating the fate of stem cells, where ROS functions as a signaling molecule to mediate stem cell commitment and differentiation. Mounting evidence highlights that stem cells exhibit a low concentration of endogenous ROS in order to maintain their stemness and pluripotency, whereas a physiological rise in ROS seems to be required to promote proper stem cell differentiation [76,77]. Accordingly, stem cells possess high levels of antioxidant enzymes to limit ROS generation and potential oxidative damage, as this may impair adult neurogenesis [44]. Because the triple tau mutations caused an increase in total mitochondrial ROS levels, the upregulation of the antioxidant enzymes SOD1, SOD2, CAT, and GPX-1 may thus serve to suppress ROS signaling, ultimately assisting in the survival and preservation of stemness. Consequently, due to the efficient detoxification, triple tau-mutant iPSCs presented lower levels of specific O_2_^•−^ and total cytosolic ROS, not only to prevent oxidative injury, but more importantly to maintain their regenerative function under pathological circumstances. Moreover, when stem cells become differentiated, their antioxidant capacity typically decreases [76], which, combined with pathology-induced mitochondrial dysfunction, may result in ROS accumulation, as observed in the above-mentioned studies. It is also worth noting that mitochondrial SIRT3 enhances the activity of multiple components of the ETC, thereby not only heightening cellular respiration, but potentially also contributing to increased mitochondrial ROS generation [78,79,80]. However, considering the variety of mechanisms stem cells have evolved to protect themselves from exceeding ROS levels, including their preferential use of glycolysis instead of OXPHOS to reduce ROS generation, the influence of the SIRT3-mediated attempt to counteract the triple tau mutations-provoked drop in energy production may hence be of minor relevance. 

Notably, changes in the cellular metabolic state and bioenergetic demand are frequently related to morphological adaptations in mitochondria, in that way reflecting their functional state. The regulation of mitochondrial dynamics therefore remains essential to sustain a healthy and uniform mitochondrial population for the proper performance of mitochondrial functions [45]. Henceforth, these dynamic changes in mitochondrial shape are coordinated via two opposing processes, fusion and fission, whereupon the balance between them ultimately modulates the degree to which mitochondria are networked. While fusion drives the mixing of mitochondrial content and as a result forms an extensively interconnected mitochondrial network, fission generates new mitochondria and separates the damaged ones from the healthy networks [81]. Concerning our findings on the mitochondrial network morphology in triple tau-mutant iPSCs, we observed a decrease in all mitochondrial shape descriptors and in mitochondrial length, suggesting an equilibrium shift towards a more fragmented mitochondrial network with shorter mitochondria. In accordance with these observations, we detected an upregulation of DRP1 and FIS1 on a transcriptional level, together with a downregulation of OPA1 in triple tau-mutant iPSCs, supporting enhanced fission as a predominating process. Intriguingly, previous findings have shown partially conflicting results regarding the impact of mutant tau on mitochondrial dynamics. Contrary to our data in iPSCs, a study in P301L tau transgenic mice (rTg4510) reported that mitochondria in hippocampal pyramidal neurons exhibit excessive mitochondrial elongation with an increase in mean mitochondrial length [82]. Nevertheless, consistent with our investigations, two in vitro studies in P301L mutant tau-transfected immortalized primary hippocampal neural cells (HT22) described a reduction in mitochondrial length [61,62]. Moreover, they found higher fission rates due to overexpressed DRP1 and FIS1 combined with downregulated OPA1, thus confirming our findings in triple tau-mutant iPSCs. Similarly, these observations were recapitulated in two in vivo models using cortical and hippocampal tissues of P301L tau transgenic mice [59,60], consequently further strengthening the notion of enhanced fission events. However, the discrepancies in the observed phenotypes among these tau models should not be surprising, given the broad variability in patients, even within members of the same family with the same mutation, or in this context, in mice. Of note, it is generally perceived that fused interconnected mitochondria are energetically more active and coupled to survival and protective functions, while fragmented and shorter mitochondria are mainly identified with a lowered activity in pathologic conditions [45]. The boost of fusion in some studies could consequently be understood as an adaptation attempt to restore and maintain a tubular mitochondrial network, aiming to reduce the pathological effects. 

Although our findings demonstrated a decreased expression level of OPA1 in triple tau-mutant iPSCs, MFN1 and MFN2 were upregulated, which are also required to mediate fusion. Despite appearing contradictory, previous studies proposed that even in the presence of functional MFNs at the outer mitochondrial membrane, the deletion of OPA1 within the inner mitochondrial membrane seems to be sufficient to promote mitochondrial fragmentation [83,84]. In addition, OPA1 acts as a master regulator of cristae structure [85], where OPA1-provoked cristae remodeling ensures the proper arrangement of the ETC respiratory complexes to support efficient OXPHOS and ATP synthesis [86]. Accordingly, the suppression of OPA1 not only represses fusion, but is also compatible with the observed bioenergetic deficits in triple tau-mutant iPSCs. 

Besides affecting the mitochondrial network morphology, the presence of the triple tau mutations also impaired the cellular distribution of mitochondria, leading to a perinuclear clustering. In line with this, similar mitochondrial localization alterations were reported in transgenic mice (rTg4510) [87] and in transfected neuroblastoma cells (SH-SY5Y) [25] bearing the P301L tau mutation. As stated above, the dynamic nature of mitochondria provides a powerful mechanism to control mitochondrial quality. Considering that fission facilitates the removal of damaged mitochondria, the increased accumulation of mitochondria around the nucleus may hence indicate their targeting for elimination through mitophagy, as the perinuclear region is associated with autophagic degradation [88]. In support of this assumption, the depolarization of the MMP, as observed in our model, usually serves as an initiator of mitophagy [89]. Nonetheless, a recent study interestingly revealed that P301L mutant tau directly impinges on mitochondrial quality control by specifically inhibiting mitophagy [90]. Given these points, one might propose that in triple tau-mutant iPSCs mitochondria undergo increased fission as an attempt to segregate dysfunctional mitochondria via mitophagy. As a result, damaged mitochondria are directed to the perinuclear area and prepared for degradation. However, since mitophagy may be compromised, an accumulation of energetically defective mitochondria could thus lead to an exacerbation of the triple tau mutation-induced mitochondrial toxicity. Nevertheless, another important point to bear in mind concerns the integral role of mitochondrial dynamics in stemness regulation. Specifically, the balance between fusion and fission has been proposed to be required for the full pluripotency of stem cells, consequently modulating various aspects of stem cell identity, fate decision, and regenerative function [44,46]. An imbalanced dynamics with excess fission is therefore implied to evoke defects in the differentiation and developmental potential of stem cells [91]. On the contrary, fusion exhibits protective effects on the mitochondrial population and appears to be crucial for development [92], as the deletion of MFN1 and MFN2 was shown to impair stem cell self-renewal in the adult hippocampus, eventually resulting in spatial learning deficits [93]. As aforementioned, García-León et al. described that the presence of the triple *MAPT* mutations caused an aberrant differentiation of iPSCs into cortical neurons and progressive cell loss [23]. For this reason, the upregulation of MFN1 and MFN2 in triple tau-mutant iPSCs could hence be interpreted as a protective compensatory mechanism to not only counteract increased fission, but ultimately to maintain the characteristics of iPSCs.

## 5. Conclusions

In summary, the present study describes for the first time the adverse impact of disease-associated tau on multiple aspects of mitochondrial function in human iPSCs, ranging from mitochondrial bioenergetics to dynamics. Hence, we provide new insights into the direct effects of mutant tau-mediated mitochondrial dysfunction in an advanced human cellular tau pathology model at early disease stages, consequently in the absence of compensatory mechanisms to slow down disease progression. Provided that mitochondrial impairments are considered to play a causative role in the pathogenesis of tauopathies and that mitochondria are perceived as fundamental regulators in stem cell function, it appears quite conceivable to envision how already primary defects in mitochondria may result in defective stem cell fate decisions, altered neurogenesis, and ultimately to a decline in neurological function. Although human iPSCs serve as an excellent platform for exploring early disease pathology, the particular influence of damaged mitochondria on their early development and differentiation has hardly been investigated. Accordingly, a better understanding of this complex relationship may thus assist in the potential prevention and treatment of tau-related neurodegeneration, in which the mitochondrial dysfunction-triggered disruption of stem cell function might contribute to their pathogenesis. In this respect, recent therapeutic strategies indicate mitochondria as viable targets for stem cell-based therapies and interventions. Particularly, the transplantation of healthy mitochondria from iPSCs with genetically corrected disease-causing mutations aims to enhance stem cell function, thereby restoring neurogenesis, which may eventually improve cognition and prevent dementia [20].

## Figures and Tables

**Figure 1 cells-12-01385-f001:**
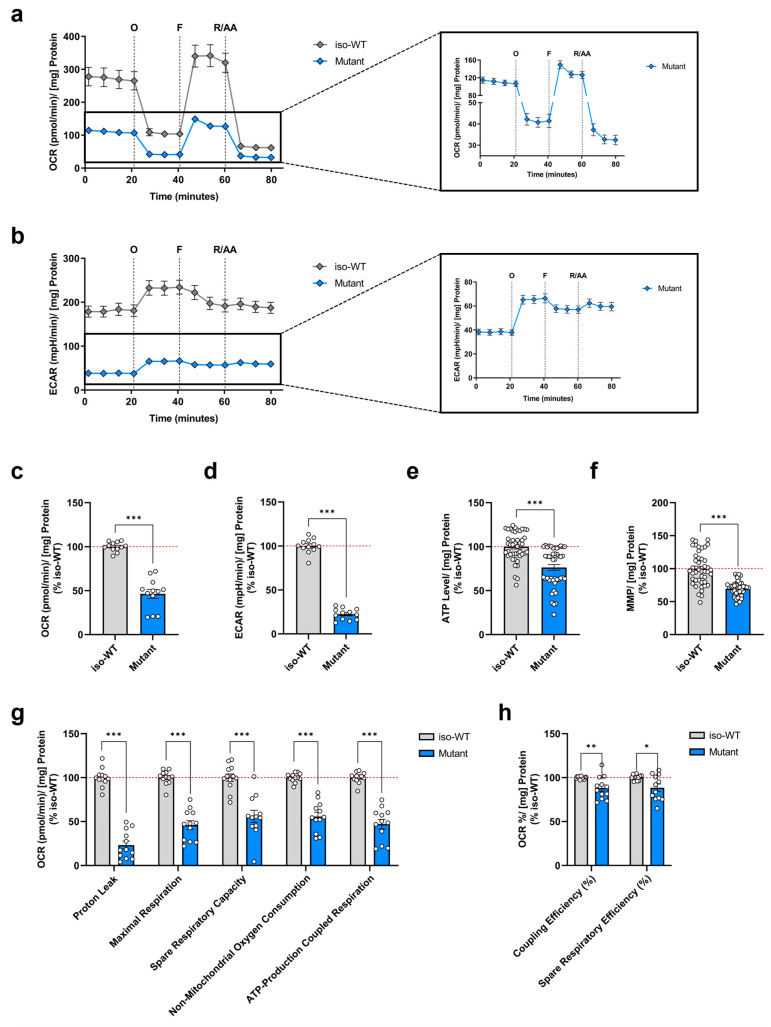
Characterization of bioenergetic deficits in triple tau-mutant iPSCs. Changes over time in (**a**) OCR and (**b**) ECAR in iso-WT and triple tau-mutant iPSCs were measured simultaneously by the sequential addition of specific respiratory modulators. Quantification of (**c**) OCR and (**d**) ECAR under basal conditions in iso-WT and triple tau-mutant iPSCs from (**a**,**b**), respectively. Determination of (**e**) relative ATP levels and (**f**) MMP in iso-WT and triple tau-mutant iPSCs. (**g**,**h**) Respiratory parameters of iso-WT and triple tau-mutant iPSCs extracted from (**a**), specifically proton leak, maximal respiration, spare respiratory capacity, non-mitochondrial oxygen consumption, ATP-production coupled respiration, coupling efficiency, and spare respiratory efficiency. Data represent the mean ± SEM of 12 replicates obtained from 4 independent Seahorse runs with n = 3 technical replicates per group for (**a**–**d**,**g**,**h**) and 4 independent experiments with n = 8–12 replicates per group for (**e**,**f**). Values were normalized on the protein content and are shown as the percentage of the iso-WT iPSCs for (**c**–**h**). Student’s unpaired *t*-test iso-WT versus mutant, * *p* < 0.05, ** *p* < 0.01, *** *p* < 0.001. AA: antimycin A; ECAR: extracellular acidification rate; F: carbonyl cyanide-p-trifluoromethoxyphenylhydrazone (FCCP); MMP: mitochondrial membrane potential; O: oligomycin; OCR: oxygen consumption rate; R: rotenone.

**Figure 2 cells-12-01385-f002:**
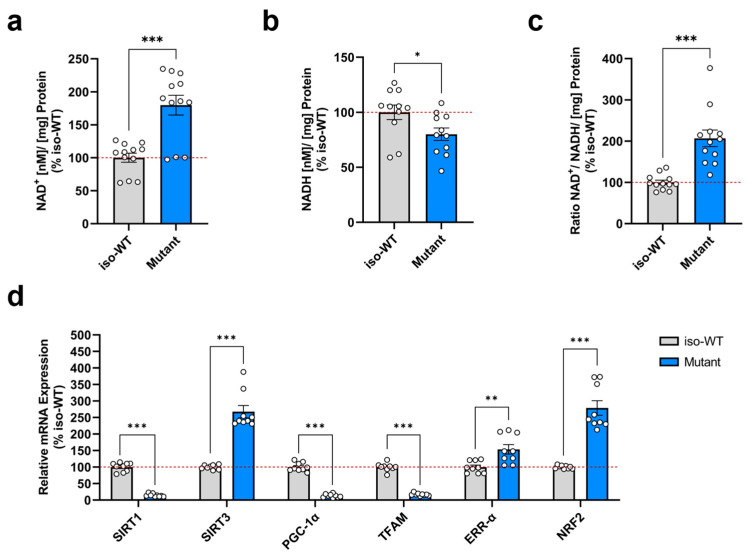
Triple tau mutations-induced effects on metabolic redox intermediates and the metabolic homeostasis control of mitochondria in iPSCs. Quantification of (**a**) total NAD^+^ concentration and (**b**) total NADH concentration of iso-WT and triple tau-mutant iPSCs. (**c**) Evaluated NAD^+^/NADH ratio of iso-WT and triple tau-mutant iPSCs. (**a**–**c**) Data represent the mean ± SEM of 4 independent experiments with n = 3 technical replicates per group. Values were normalized on the protein content and are shown as the percentage of the iso-WT iPSCs. (**d**) Changes in relative mRNA expression of genes involved in mitochondrial biogenesis and metabolic control in iso-WT and triple tau-mutant iPSCs, namely SIRT1, SIRT3, PGC-1α, TFAM, ERR-α, and NRF2. Data represent the mean ± SEM of 3 independent experiments with n = 3 replicates per group. Gene expression was related to GAPDH and values are shown as the percentage of the iso-WT iPSCs. Student’s unpaired *t*-test iso-WT versus mutant, * *p* < 0.05, ** *p* < 0.01, *** *p* < 0.001. ERR-α: estrogen-related receptor α; GAPDH: glyceraldehyde-3-phosphate dehydrogenase; NAD: nicotinamide adenine dinucleotide; NRF2: nuclear respiratory factor 2; PGC-1α: peroxisome proliferator-activated receptor-γ co-activator-1α; SIRT1: sirtuin 1; SIRT3: sirtuin 3; TFAM: mitochondrial transcription factor A.

**Figure 3 cells-12-01385-f003:**
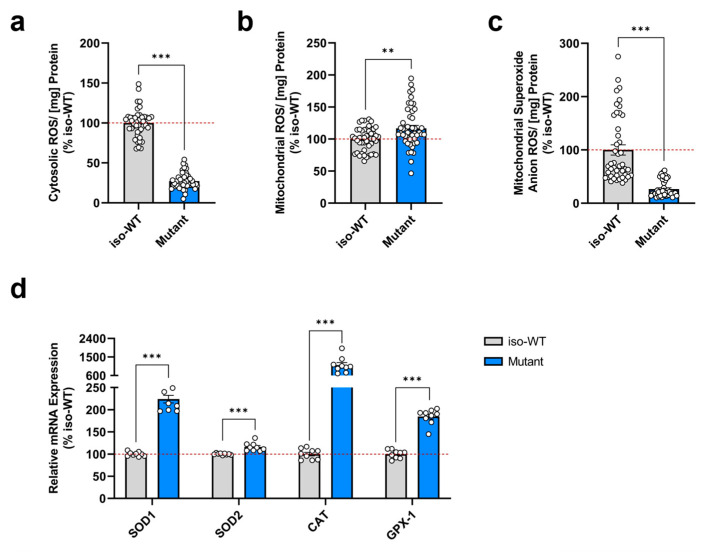
Triple tau mutations influence the cellular redox state and the antioxidative defense system in iPSCs. Assessment of (**a**) total cytosolic ROS levels, (**b**) total mitochondrial ROS levels, and (**c**) specific mitochondrial superoxide anion radical levels in iso-WT and triple tau-mutant iPSCs. (**a**–**c**) Data represent the mean ± SEM of 4 independent experiments with n = 8–12 replicates per group. Values were normalized on the protein content and are shown as the percentage of the iso-WT iPSCs. (**d**) Changes in relative mRNA expression of genes involved in the antioxidative defense system in iso-WT and triple tau-mutant iPSCs, specifically SOD1, SOD2, CAT, and GPX-1. Data represent the mean ± SEM of 3 independent experiments with n = 3 replicates per group. Gene expression was related to GAPDH and values are shown as the percentage of the iso-WT iPSCs. Student’s unpaired *t*-test iso-WT versus mutant, ** *p* < 0.01, *** *p* < 0.001. CAT: catalase; GAPDH: glyceraldehyde-3-phosphate dehydrogenase; GPX-1: glutathione peroxidase 1; ROS: reactive oxygen species; SOD1: superoxide dismutase 1; SOD2: superoxide dismutase 2.

**Figure 4 cells-12-01385-f004:**
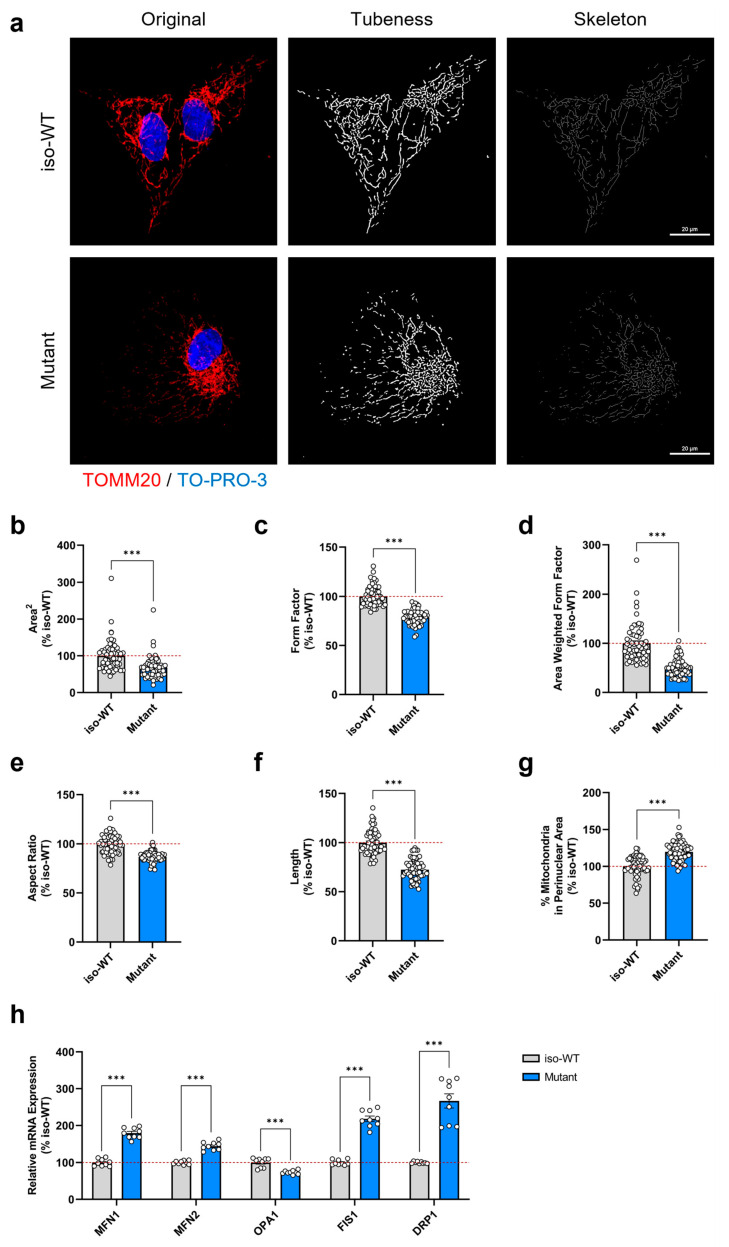
Triple tau mutations-evoked impact on mitochondrial network morphology, distribution, and dynamics in iPSCs. (**a**) Representative microscopy images (z-projections) of the mitochondrial network immuno-stained with the TOMM20 antibody (mitochondrial marker) and with TO-PRO-3 (nuclear staining) in iso-WT and triple tau-mutant iPSCs (63× magnification, scale bar = 20 µm). The left panels (original) show the merged images of the mitochondrial network (in red) and the nucleus (in blue). The center panels (tubeness) display the mitochondrial network (in gray) after image processing using the morphometry macro in FIJI. The right panels (skeleton) present the mitochondrial network (in gray) after further image processing in FIJI using the skeletonize function. (**b**–**f**) Quantification of mitochondrial network morphology metrics after image processing using the morphometry macro in FIJI for the evaluation of (**b**) area^2^ (average mitochondrial size), (**c**) form factor (mitochondrial elongation), (**d**) area-weighted form factor (a variant of the form factor with a bias towards larger mitochondria), (**e**) aspect ratio (ratio of the major and minor axis), and (**f**) length (mitochondria) in iso-WT and triple tau-mutant iPSCs. (**g**) Evaluation of the percentage of mitochondria in the perinuclear area using the radial fluorescence plugin in FIJI. (**b**–**g**) Data represent the mean ± SEM of 4 independent experiments with n = 15 images per group. Values are shown as the percentage of the iso-WT iPSCs. (**h**) Changes in relative mRNA expression of genes involved in mitochondrial fusion and fission in iso-WT and triple tau-mutant iPSCs, namely MFN1, MFN2, OPA1, FIS1, and DRP1. Data represent the mean ± SEM of 3 independent experiments with n = 3 replicates per group. Gene expression was related to GAPDH and values are shown as the percentage of the iso-WT iPSCs. Student’s unpaired *t*-test iso-WT versus mutant, *** *p* < 0.001. DRP1: dynamin-related protein 1; FIS1: mitochondrial fission 1; GAPDH: glyceraldehyde-3-phosphate dehydrogenase; MFN1: mitofusin 1; MFN2: mitofusin 2; OPA1: optic atrophy 1; TOMM20: translocase of the outer mitochondrial membrane complex subunit 20.

**Figure 5 cells-12-01385-f005:**
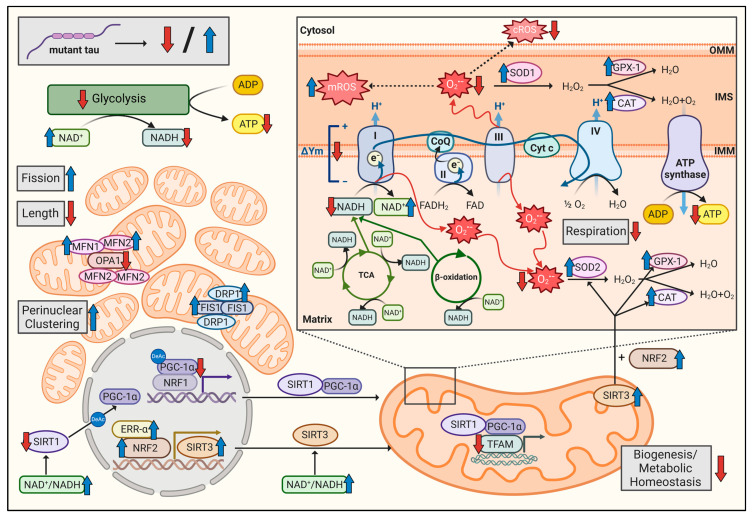
Schematic representation of the triple tau mutations-evoked impairments of multiple aspects of mitochondrial function in iPSCs. Based on the findings obtained in the present study, the figure illustrates the triple tau mutations-induced alterations in mitochondrial function. Red arrows indicate a decrease and blue arrows an increase in the depicted parameters for triple tau-mutant iPSCs. ADP: adenosine diphosphate; ATP: adenosine triphosphate; CAT: catalase; cROS: cytosolic ROS; CoQ: coenzyme Q10; Cyt c: cytochrome c; DRP1: dynamin-related protein 1; ERR-α: estrogen-related receptor α; FIS1: mitochondrial fission 1; GPX-1: glutathione peroxidase 1; IMM: inner mitochondrial membrane; IMS: intermembrane space; mROS: mitochondrial ROS; MFN1: mitofusin 1; MFN2: mitofusin 2; NAD: nicotinamide adenine dinucleotide; NRF1: nuclear respiratory factor 1; NRF2: nuclear respiratory factor 2; OMM: outer mitochondrial membrane; OPA1: optic atrophy 1; O_2_^•−^: mitochondrial superoxide anion radicals; ROS: reactive oxygen species; SIRT1: sirtuin 1; SIRT3: sirtuin 3; SOD1: superoxide dismutase 1; SOD2: superoxide dismutase 2; TCA: tricarboxylic acid cycle; TFAM: mitochondrial transcription factor A; ΔΨm: mitochondrial membrane potential; I: complex I; II: complex II; III: complex III; IV: complex IV. Created with BioRender.com.

**Table 1 cells-12-01385-t001:** Calculations of Seahorse assay parameters.

Assay Parameter	Equation	Depicted in Figure
Non-Mitochondrial Oxygen Consumption	Minimum rate measurement after rotenone/antimycin A injection	1g
Basal Respiration	(Last rate measurement before first injection)—(Non-Mitochondrial Oxygen Consumption)	
Maximal Respiration	(Maximum rate measurement after FCCP injection)—(Non-Mitochondrial Oxygen Consumption)	1g
Proton Leak	(Minimum rate measurement after oligomycin injection)—(Non-Mitochondrial Oxygen Consumption)	1g
ATP-Production Coupled Respiration	(Last rate measurement before oligomycin injection)—(Minimum rate measurement after oligomycin injection)	1g
Spare Respiratory Capacity	(Maximal Respiration)—(Basal Respiration)	1g
Spare Respiratory Efficiency	(Maximal Respiration)/(Basal Respiration) × 100%	1h
Coupling Efficiency	(ATP-Production Coupled Respiration)/(Basal Respiration) × 100%	1h

**Table 2 cells-12-01385-t002:** Primer sequences.

Target Gene	Forward Primer	Reverse Primer
CAT	5′-AGATGCAGCACTGGAAGGAG-3′	5′-GCATGCACAACTCTCTCAGG-3′
DRP1	5′-TCACGAGACAAGTCTTCTAAAG-3′	5′-CCTCCAGATGCAACCTTG-3′
ERRα	5′-TAGTTACCTTGGGCACTGGG-3′	5′-GTTTGTGGACAGGCTGTGAG-3′
FIS1	5′-GCTCAAGGAATACGAGAAGG-3′	5′-AGTCCATCTTTCTTCATGGC-3′
GAPDH	5′-CATGGTTTACATGTTCCAATATGA-3′	5′-GGATCTCGCTCCTGGAAG-3′
GPX1	5′-GTGCTCGGCTTCCCGTGCAAC-3′	5′-CTCGAAGAGCATGAAGTTGGGC-3′
MFN1	5′-CAGTCACCAAGTAAAACAACAAA-3′	5′-GGGTAATCTAGCAATTTCTTCTTC-3′
MFN2	5′-CAGGATTCAGAAAGCCCAG-3′	5′-GATGCACTCCTCAAATCTCC-3′
NRF2	5′-CCAGCACATCCAGTCAGAA-3′	5′-AGCCGAAGAAACCTCATTGT-3′
OPA1	5′-CCAGGTGTGATTAATACTGTGA-3′	5′-CCATCTTGAATACACAGTATGATG-3′
PGC1-α	5′-TCCTCTTCAAGATCCTGCTATTAC-3′	5′-TCTCAGACTCTCGCTTCTCATA-3′
SIRT1	5′--3′GTAGGCGGCTTGATGGTAAT-3	5′-GGGTTCTTCTAAACTTGGACTCT-3′
SIRT3	5’-CATGAGCTGCAGTGACTGGT-3’	5’-GAGCTTGCCGTTCAACTAGG-3′
SOD1	5′-GCTGGTTTGCGTCGTAGTCT-3′	5′-ATGCAGGCCTTCAGTCAGTC-3′
SOD2	5′-AAGCACCACGCGGCCTACG-3′	5′-CCATTGAACTTCAGTGCAGGCTG-3′
TFAM	5′-CAAGTTGTCCAAAGAAACCTGTAA-3′	5′-GCCACTCCGCCCTATAAG-3′

Abbreviations: CAT: catalase; DRP1: dynamin-related protein 1; ERR-α: estrogen-related receptor α; FIS1: mitochondrial fission 1; GAPDH: glyceraldehyde-3-phosphate dehydrogenase; GPX-1: glutathione peroxidase 1; MFN1: mitofusin 1; MFN2: mitofusin 2; NRF2: nuclear respiratory factor 2; OPA1: optic atrophy 1; PGC1-α: peroxisome proliferator-activated receptor-γ co-activator-1α; SIRT1: sirtuin 1; SIRT3: sirtuin 3; SOD1: superoxide dismutase 1; SOD2: superoxide dismutase 2; TFAM: mitochondrial transcription factor A.

## Data Availability

The data presented in this study are available on request from the corresponding author.

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
