# Peer review of "Genetically Engineered Triple MAPT-Mutant Human-Induced Pluripotent Stem Cells (N279K, P301L, and E10+16 Mutations) Exhibit Impairments in Mitochondrial Bioenergetics and Dynamics"

_cells, 2023, doi:10.3390/cells12101385_

Round 1

Reviewer 1 Report

Szabo and colleagues have characterized the mitochondrial function and mitochondrial dynamics of a triple MAPT mutant iPSC line. The authors previously showed the outcome of the N279K, P301L and E10+16 triple mutation on tau biochemistry, oxidative stress, glutamate signalling and electrophysiology in cortical neurons derived from the same iPSCs (Garcia-Leon et al). The topic is interesting and the manuscript is well written. However, some aspects of the methodology, statistics and figures are unclear and therefore I have some questions and comments for the authors.

General concerns, questions and comments:

·         It is unclear why the study was carried out using a non-isogenic wild type iPSC control derived from a healthy 24 year old male donor when the isogenic control ChiPS6b healthy donor-derived WT cells (purchased from Takara Bio Inc.) exists. Considering the large amount of mitochondrial heterogeneity between healthy individuals, can the authors justify this comparison between a single wild type iPSC line and the triple MAPT mutant?

·         Due to high mitochondrial variability between cells and individuals I am concerned about hyper-normalisation of the data. All of the data in the manuscript except Figure 1 A and 1B is expressed as % WT and the WT shows SEM error bars. Can the authors explain how the SEM error for the WT % control is calculated? Since the wild type control here does not have the same genetic background as the MAPT mutant line, one could argue that the unseen variability between independent experiments and the lines may reduce the ability to draw any strong conclusions? Could the authors comment on this point? And/or show examples of non-normalised data?

·         The interest of the readers and scientific significance of this mitochondrial characterisation weighs on the MAPT triple mutant, which has implications in several neurodegenerative diseases. I agree that understanding the mitochondrial biology underlying these diseases is important. Therefore, could the authors offer their reasoning why iPSCs were used as the model system instead of neurons or neural cells such as NPCs? This should be discussed in the manuscript since iPSCs, mitotic neural cells and mature neurons have different mitochondrial metabolism.

Specific concerns, questions and comments:

·         Why are there no error bars in figures 1A and 1B (Mutant)?

·         In figure 1G, the spare respiratory capacity is highly significantly reduced in the MAPT mutant compared to WT. In the figure 1A, the spare respiratory capacity does not look too different between WT and mutant. How many independent experiments were performed and shown in Figure1?

·         If there is a drastic reduction of basal OCR and ECAR in mutant iPSCs, what is driving energy production? If there is an overall drastic reduction of mitochondrial respiration and glycolysis, would one not expect a growth or survival defect? Since ATP levels are only ~20% lower, this needs to be clarified. On the topic, could the authors explain the difference in their calculations between Figure 1G and 1H?

·         For Figure 1. The statistics in the figure legend are not clear. Am I looking at the statistics for n=4 independent experiments? What n number was used to calculate the statistics? I am assuming that the technical replicates here are multiple wells of iPSCs seeded in the independent seahorse assay run? Or something else, seahorse runs?

·         Can the authors justify the parametric T test? Was the data normally distributed?

·         In the discussion it is mentioned that iPSCs primarily utilize glycolysis. In this study glycolysis and mitochondrial respiration are shown to be dramatically reduced at baseline. Experiments to measure MAPT mutant iPSC cell viability would support this hypothesis along with some of the key experiments measuring mitochondrial dysfunction in derived NPCs or neurons would be more convincing. Since the metabolic effects are so significant in the iPSCs, I wonder how this relates to their capability to differentiate into cortical neurons (shown in Garcia-Leon et al).

With the focus on mitochondria, this manuscript describes the same mutant cells as in Garcia-leon et al. but as iPSCs but as compared to a non-isogenic WT line. Without an isogenic control or several healthy iPSC lines to account for the variability between individuals, a few key experiments are needed to convince that such striking hyper-normalised mitochondrial changes are indeed relevant.

Reviewer 2 Report

The authors characterize the mitochondrial function in engineered human induced pluripotent stem cells by combining three microtubule-associated protein TAU mutations. IPSC are powerful translational models for tauopathies that allow the explore the disease phenotypic alterations over the development process, iPSC into neurons. Mitochondrial dysfunction is a common hallmark of several neurodegenerative disorders. In most cases, the link mechanism underlying mitochondria dysfunction and disease-causing mutations still needs to be clarified. More important is urgent to understand if the mitochondria dysfunction is a late consequence of these mutations or an early event that contributes to disease phenotype and progression.

I believe the data is useful to readers and experts in the field of tauopathies since there are still no effective disease-modifying therapies and mitochondrial can be a potential therapeutic target.

The article is well organized and easy to understand with all the sections well-developed.

The methods are well explained.

The results are clearly explained.

I have some comments for the authors: 

1) Why did the authors choose iPSC as models instead of neural progenitors or differentiated neurons that resemble the disease phenotype more accurately?

As the authors refer in the discussion the mitochondria metabolism/morphology changes dramatically over differentiation and neurons depend more on OXPHOS for ATP production than iPSC.

2) Do the authors know the mechanism linking these tau mutations to mitochondria? Is there a direct relation interaction or the effect is indirect?

2) The authors performed the seahorse assay to evaluate mitochondrial respiration and analyze OCR and ECAR by using the XF Mito Stress Test. Since the glycolytic pathway is preferentially used by iPSC the authors should have performed the Glycolysis Stress Test to profile the glycolysis.

3) In panel 1 the data shown in Figures C and G for OCR and basal respiration seem to be duplicated. For Figure H the authors should describe in the methods how they calculated coupling efficiency, and spare respiratory efficiency.

All the parameters analyzed are decreased in triple tau-mutant iPSCs mitochondria, but I would expect the mitochondria to be more “leaky” and have increased proton leak. Can the authors explain why the values are so reduced compared to controls?

4) Line 391 “Hence, the calculated NAD+/ NADH ratio was significantly augmented in triple tau-mutant iPSCs versus WT iPSCs (Figure 2c), suggesting a higher oxidative status.” Can the authors define more precisely the “higher oxidative status”?

5) In Figure 3 is interesting that mitochondrial ROS is increased but mitochondrial superoxide anion is decreased. I suggest the authors measure the mitochondrial H2O2 production with a MitoPY probe to clarify these results.

Round 2

Reviewer 1 Report

Thank you for explaining the statistics, experimental setup and source of the donor cells. I see that the information is clearly presented in the methods section and in the figure legends.

The figures are also much improved.

I am satisfied with the improved manuscript.

Reviewer 2 Report

The authors addressed most of my concerns in the revised version. I have no more comments and I recommend the manuscript be accepted in the current form.